# Exploiting Implicit Rigidity Constraints via Weight-Sharing Aggregation for Scene Flow Estimation from Point Clouds

## Abstract

Scene flow estimation, which predicts the 3D motion of scene points from point clouds, is a core task in autonomous driving and many other 3D vision applications. Existing methods either suffer from structure distortion due to ignorance of rigid motion consistency or require explicit pose estimation and 3D object segmentation. Errors of estimated poses and segmented objects would yield inaccurate rigidity constraints and in turn mislead scene flow estimation. In this paper, we propose a novel weight-sharing aggregation (WSA) method for feature and scene flow up-sampling. WSA does not rely on estimated poses and segmented objects, and can implicitly enforce rigidity constraints to avoid structure distortion in scene flow estimation. To further exploit geometric information and preserve local structure, we design a deformation degree module aim to keep the local region invariance. We modify the PointPWC-Net and integrate the proposed WSA and deformation degree module into the enhanced PointPWC-Net to derive an end-to-end scene flow estimation network, called WSAFlowNet. Extensive experimental results on the FlyingThings3D (Mayer et al., 2016) and KITTI (Menze et al., 2018) datasets demonstrate that our WSAFlowNet achieves the state-of-the-art performance and outperforms previous methods by a large margin. We will release the source code of WSAFlowNet upon the publicity of the paper.

## 1 Introduction

Estimating the 3D motion of scene points from two consecutive frames, known as scene flow estimation, is vital to many 3D applications including autonomous driving (Zhai et al., 2020; Behl et al., 2019; Pontes et al., 2020; Deng & Zakhor, 2023). Traditional methods usually estimate scene flow from RGB or RGB-D images (Menze & Geiger, 2015; Pons et al., 2007; Chi et al., 2021; Wedel et al., 2011; Quiroga et al., 2014). Recently, due to the increasing application of 3D sensors such as LiDAR, directly estimating scene flow from 3D point clouds has attracted a lot of interests.

In recent years, with the evolution of deep learning, neural networks have become a popular approach for handling point cloud data. For instance, FlowNet3D (Liu et al., 2019) designs an end-to-end scene flow estimation network based on PointNet++ and introduces a flow embedding layer to encode 3D motion between the source and target point clouds. Similarly, PointPWC-Net (Wu et al., 2020) adopts PointConv (Wu et al., 2019) as the convolution operation, and proposes a learnable patch-to-patch cost volume and a coarse-to-fine strategy to improve the accuracy, in particular for points with large displacements. However, the coarse-to-fine strategy used in PointPWC-Net could also lead to error accumulation in the early stages. To address this problem, PV-RAFT (Wei et al., 2021) leverages a gated recurrent unit (GRU (Teed & Deng, 2020) ) based optimization architecture. Specifically, PV-RAFT builds the point-voxel correlation fields to capture both local and long-range motion and then iteratively optimizes the estimated motion via GRU updaters. Despite the success of these works, they ignore the fact that the scene flow for points of the same rigid object should conform to the same geometric transformation, i.e., rigid motion consistency. As a result, these methods could suffer from rigid structure distortion in the scene flow estimation.

To solve the above mentioned problem, we propose a network that utilizes implicit rigidity constraints based on the coarse-to-fine architecture. We focus on the internal relations within the point's

neighborhood $N(p_i)$. First, we propose a weight-sharing aggregation approach which is utilized in the upsampling layer and implicitly enforces the rigidity constraints to avoid structure distortion in scene flow estimation. Importantly, our method does not require explicit rigid object clustering/segmentation or pose estimation, which helps to avoid error interference that may arise from combining multiple tasks. The key idea of weight-sharing aggregation is grounded in our mathematical analysis. For a given 3D point $p_i$ and its neighboring points $N(p_i)$, using identical weights to aggregate the point coordinate, along with their features and the scene flow of $N(p_i)$ can enforce consistent motion implicitly among the points in $N(p_i)$.

Furthermore, we propose a deformation degree module which measures the deformation degree of local structure between the reference points $N(p_i)$ and the warped points $N(p_i^w)$. It is constructed using the source point cloud and predicted scene flow, to further enhance local structure consistency. The deformation degree module serves as an additional input for our estimator providing supplementary geometric information. We build a scene flow estimation network, named WSAFlowNet, by integrating our weight-sharing aggregation and deformation degree module into an enhanced version of PointPWC-Net (Wu et al., 2020). Extensive experimental results demonstrate the effectiveness and generalization capability of our method. Our method surpasses the current state-of-the-art methods Bi-PointFlowNet (Cheng & Ko, 2022). According to the EPE3D metric, we outperform Bi-PointFlowNet by $14.6\%$ on the FlyingThings3D (Mayer et al., 2016) dataset, and by $7.6\%$ on the KITTI (Menze et al., 2018) dataset.

In summary, this paper makes three contributions:

1. A novel weight-sharing aggregation approach for upsampling which implicitly enforces rigidity constraints by using identical weights for aggregation of point coordinates, scene flow and features. We mathematically and experimentally prove the feasibility of our approach. In addition, our approach does not require explicit pose estimation and/or 3D object segmentation.

2. A deformation degree module which captures the discrepancy in local structure between the source point cloud and the warped point cloud. This module helps to preserve the local structure of rigid objects during scene flow estimation.

3. An effective scene flow estimation network, named WSAFlowNet, which integrates these two approaches to achieve the state-of-the-art performance on the FlyingThings3D and KITTI datasets.

## 2 RELATED WORK

**Scene flow estimation.** Several recent methods have achieved impressive performance on the scene flow estimation task (Wu et al., 2020; Battrawy et al., 2022; Gu et al., 2019b; Wang et al., 2020; 2021; Gu et al., 2022). PointPWC-Net (Wu et al., 2020) introduces a coarse-to-fine network architecture to estimate scene flow. It can capture large motions without enlarging the scope of search. Additionally, it proposes a novel patch-to-patch cost volume to encode point motions effectively. RMS-FlowNet (Battrawy et al., 2022), on the other hand, designs a Patch-to-Dilated-Patch flow embedding block that, in conjunction with Random-sampling, allows for operation on large-scale point clouds. The methods mentioned above only use unidirectional features, which can result in insufficient information. To address this limitation, Bi-PointFlowNet (Cheng & Ko, 2022) introduces bidirectional flow embedding layers to extract correlations both forward and backward. 3DFlow (Wang et al., 2022) proposes an all-to-all flow embedding layer that can capture distant points and combine them with backward reliability validation. However, this pointwise scene flow estimation method does not make full use of structural information during movement. Consequently, they may encounter structure distortion in some challenging scenes, which is often caused by the sparsity of point clouds.

**Rigidity constraints.** A straightforward solution to prevent structure distortion is to explicitly estimate geometric transformations of rigid objects in a scene and enforce rigid motion consistency on estimated scene flows. To this end, Gojcic et al. (Gojcic et al., 2021) categorize scene points into the foreground clusters, which consist of several rigid objects, and the background cluster. They then compute the ego-motion and the geometric transformation of each object between the source and target point clouds via (Kabsch, 1976; Yew & Lee, 2020; Cuturi, 2013). Next, they enforce the estimated scene flows of the same object to conform to the same geometric transformation. Dong et al. (Dong et al., 2022) also adopt a similar approach, where they generate an abstraction mask

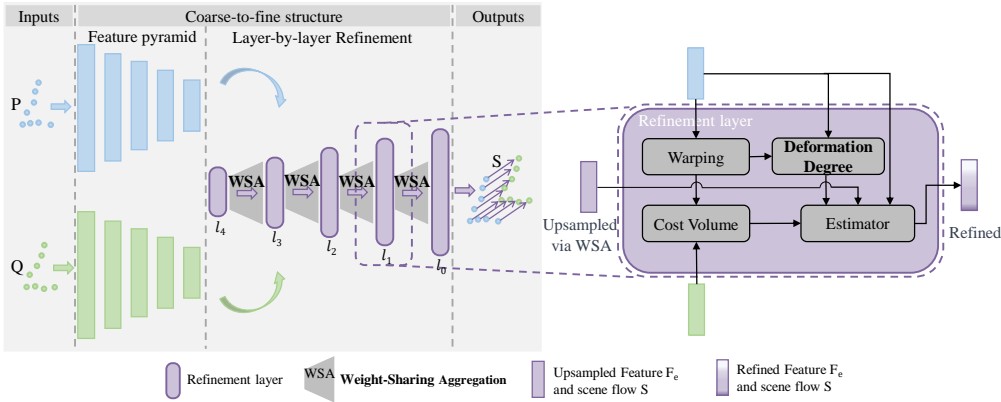

Figure 1: **WSAFlowNet overview.** On the left side, the entire pipeline of the coarse-to-fine network is illustrated. The inputs of the network are two consecutive sets of point clouds $P, Q$, and outputs the scene flows corresponding to the $P$. The process primarily consists of two main stages: feature abstraction and the refinement of scene flow estimation. Between adjacent refinement layers, we incorporate the **weight-sharing aggregation** (Fig. 2) module for the upsampling of feature and scene flow. The right side highlights **the refinement layer**. Our designed **deformation degree module** (Fig. 3) maintains local structure.

for each input source point cloud using a pre-trained segmentation network (Gojcic et al., 2021) and the DBSCAN clustering algorithm (Ester et al., 1996). They then estimate poses for each object using (Kabsch, 1976). After that, direct multi-body rigidity constraints are computed based on the abstraction mask and poses. These constraints are then integrated into the recurrent neural network to alleviate structure distortion in scene flow estimation. To alleviate inevitable errors in 3D object segmentation, HCRF-Flow (Li et al., 2021) treats spatially neighboring points as a rigid object and employs a conditional random fields (CRF (Tseng et al., 2005)) to enforce local smoothness and rigid motion consistency. However, all the aforementioned methods require explicit object pose estimation, which is also very challenging. Inevitably errors in pose estimation can lead to inaccurate constraints and in turn mislead scene flow estimation.

## 3 METHOD

### 3.1 OVERVIEW

The inputs of our network are two consecutive point clouds $P = \left\{ p_i \in \mathbb{R}^3 \right\}_{i=1}^{N_1}$ at timestamp t and $Q = \left\{ q_i \in \mathbb{R}^3 \right\}_{j=1}^{N_2}$ at timestamp t+1. Our objective is to estimate the scene flow $S = \left\{ s_i \in \mathbb{R}^3 \right\}_{i=1}^{N_1}$ for every $p_i \in P$. Due to the inherent sparsity of point cloud data, there isn't a direct one-to-one correspondence between sets $P$ and $Q$. Thus, the relationship between $P$, $Q$, and $S$ can only be expressed approximately, which is represented by $P + S \approx Q$. Furthermore, we assume that the majority of local regions satisfy the approximately rigid assumption as described in HCRF-Flow (Li et al., 2021) and Man et al. (Man & Vision, 1982). Consequently, the movement of point clusters within such regions adhere to a consistent transformation, characterized by the rotation matrix $R \in SO(3)$ and the translation vector $t \in \mathbb{R}^3$.

As shown in Fig. 1, We adopt the coarse-to-fine structure PointPWC-Net (Wu et al., 2020) as the baseline. First, we construct feature pyramid for point clouds $P$ and $Q$. Next, based on the extracted feature, we conduct the scene flow estimation in the coarsest layer and then operate the layer-by-layer refinement to the finest layer. Between adjacent layers, we propose a weight-sharing aggregation module for upsampling feature and scene flow. The upsampled scene flow will be propagated by warping, which will enable the adjustment of the search center for matching. The upsampled feature will be fed into the scene flow estimator to supply movement information. Notably, at each layer, we systematically execute the following operations in sequence: the warping process, the

construction of cost volume, the deformation degree module and the scene flow estimator. Among these, the deformation degree module we proposed is utilized to preserve the geometric structure.

## 3.2 WEIGHT-SHARING AGGREGATION MODULE

In the scene flow estimation task, the sparsity of point cloud data often leads to poor correspondence. This lack of correspondence results in matching errors, causing the warped point clouds to undergo structural deformation. To alleviate the problem of mismatch, we introduce a weight-sharing aggregation module that leverages implicit rigidity constraints during the upsampling process.

In the upsampling process, weight aggregation is a common implementation that involves assigning weights to neighboring points in a coarser layer to interpolate central point information. Other methods don't notice the relationship between different variables and don't involve point coordinates upsampling. In this paper, we implicitly utilize rigidity constraints by considering the relationship between coordinates, features, and scene flow.

According to rigidity constraints (points on the same object have the same motion $(R, t)$) and the definition of scene flow, the scene flow aggregation process corresponding to each point should be consistent with the coordinate aggregation process of the point, that is, weight-sharing, as shown in Eq. (1-6). In addition, feature is an intermediate quantity closely related to scene flow and point coordinates, so we extend the weight sharing to the feature level, namely Eq. (7). In specific implementation, WSA module is applied in the upsampling process of refinement stage. It contains the condition that points on the same object conform to the same motion and implicitly enforces rigidity constraints, which reduces the impact of mismatch.

**The weight-sharing aggregation constraints.** In the local rigid region where the points inside have the same movement $(R, t)$, if aggregated weight $\alpha_k$ satisfies both Eq. (1) and Eq. (2), then it can derive Eq. (3) related to the aggregation of scene flow.

**Conditions:**

$$\sum_{k=1}^{K} \alpha_k = 1 \tag{1}$$

$$\sum_{k=1}^{K} \alpha_k p_k = p_i \tag{2}$$

**Conclusion:**

$$\sum_{k=1}^{K} \alpha_k s_k = s_i \tag{3}$$

Where $p_k \in N(p_i)$, $N(p_i)$ is defined as the group of K nearest neighbors of $p_i$, determined by their distance from $p_i$. The constant $K$ can be set to any value as required, which represents the size of the neighborhood. $s_i$ is the predicted scene flow of point $p_i$.

**Proof:**

Utilize $R, t$ to denote the motion of $p_k$ as Eq. (4).

$$T(p_k) = R p_k + t \tag{4}$$

According to the definition of scene flow, aggregation of $S$ is explicitly expressed as:

$$s_k = T(p_k) - p_k = (R - I)p_k + t \tag{5}$$

$$\begin{aligned}
\sum_{k=1}^{K} \alpha_k s_k &= \sum_{k=1}^{K} \alpha_k \{(R - I)p_k + t\} \\
&= (R - I)\sum_{k=1}^{K} \alpha_k p_k + t \sum_{k=1}^{K} \alpha_k \\
&= (R - I)p_i + t \\
&= s_i
\end{aligned} \tag{6}$$

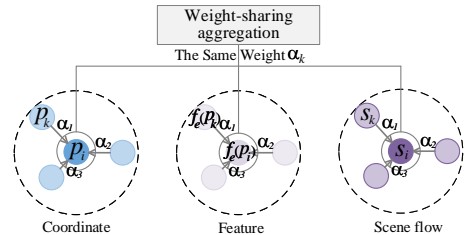

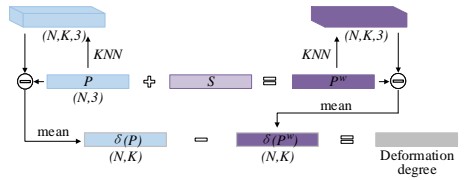

Figure 2: **Weight-sharing aggregation module.** During the upsampling process, the weights among point coordinates, features and scene flow are consistent. $f_e\,(p_i)$ and $s_i$ are the feature and scene flow corresponding to $p_i$.

Figure 3: **Deformation degree module.** Calculation of the neighborhood relationship among the point and the neighborhood points is illustrated in the upper part. Then we compare the neighborhood relationship between the source point cloud's $\delta(P)$ and the warped point cloud's $\delta(P^w)$.

We infer that the feature, intermediate variable between point cloud and scene flow, should have a matching consistency relationship.

Corresponding equation goes here:

$$\sum_{k=1}^{K} \alpha_k f_e\,(p_k) = f_e\,(p_i) \tag{7}$$

where $f_e\,(p_i)$ denotes the output feature of scene flow estimator for point $p_i$.

### 3.3 DEFORMATION DEGREE MODULE

In addition to using $(R, t)$ to make rigidity constraints according to the definition of scene flow, another way to implement rigidity constraints is to maintain local structure invariance.

We propose a novel module named deformation degree module to maintain local rigid structure invariance, which provides a stronger loss constraint. Firstly, we construct a variable $\delta$ to represent the local structure, calculating the distance between the central point and the $K$ nearest neighborhood points in the local rigid area. We maintain the dimension of $K$ to enforce a one-to-one structural relationship between the central point and its neighborhood points. Secondly, we calculate the difference $\delta_{DD}$ between $\delta(P)$ and $\delta(P^w)$. As a priori knowledge, in a rigid transformation, the geometric structure of an object remains unchanged. This ensures that the distance between any two points within a local rigid region stays constant, irrespective of the object's motion within the scene. Therefore, we enforce a loss to make the difference $\delta_{DD}$ converge to 0 to preserve the local structure and we input the $\delta_{DD}$ into scene flow estimator to supplement the geometric information. In this way, we utilize the stronger prior information to reducing rigid objects' local deformation.

$$p_i^w = p_i + s_i \tag{8}$$

$$\delta(P) = \left\{ \frac{1}{C}\,|p_k - p_i| \mid p_k \in N\,(p_i) \right\} \tag{9}$$

$$\delta(P^w) = \left\{ \frac{1}{C}\,|p_k^w - p_i^w| \mid p_k^w \in N\,(p_i^w) \right\} \tag{10}$$

$$\delta_{DD} = |\delta(P) - \delta(P^w)| \tag{11}$$

where $p_i^w$ represents the warped point. $N\,(p_i)$, $N\,(p_i^w)$ represents the neighborhood centered on $p_i$ and $p_i^w$ respectively. $C$ indicates the number of channels.

### 3.4 NETWORK ARCHITECTURE

We adopt PointPWC-Net (Wu et al., 2020) as our baseline, equip it with our proposed module, and modify the network structure. The pipeline (Fig. 1) is a coarse-to-fine structure. It first estimates the scene flow at the lowest resolution and then refine several times to the highest resolution.

**Feature pyramid.** We adopt the $set\_conv$ layer proposed by FlowNet3D (Liu et al., 2019) to encode feature and downsample $l-1$ layer to $l$ layer by Farthest Point Sampling. Our pyramid structure has five levels $\{l_0 - l_4\}$ by adding a layer for estimating at the roughest resolution ($1/128$ of the input scale), so that doesn't cause a lot of memory consumption. For the $l_3$ level, adding a rougher layer to provide scene flow initialization for cost volume construction is better than operating it directly.

**WSA in the upsampling layer.** Upsampling layer is a process of interpolating the coordinates, features, and scene flow from a coarse layer to a finer layer. Our WSA module involves using a consistent weighted aggregation method for all three components, and we will focus on the features as an example. For each point $p_i^{l-1}$ in the finer level $l-1$, we select its $K$ nearest neighbors $p_k^l$ in the coarser level $l$, denoted by $p_k^l \in N\left(p_i^{l-1}\right)$. The upsampling process for features are formulated as follows.

$$\alpha_k = \text{softmax}\left(\text{mean}\left(\text{MLP}\left(\left[p_k^l - p_i^{l-1}, f_e\left(p_k^l\right)\right]\right)\right)\right) \tag{12}$$

$$f_e\left(p_i^{l-1}\right) = \sum_{k=1}^{K} \alpha_k f_e\left(p_k^l\right) \tag{13}$$

where $[\cdot, \cdot]$ indicates concatenation operation. $\text{mean}\left(\right)$ indicates operating the average operation in the channel dimension. $f_e\left(p_i^l\right)$ denotes the output feature of scene flow estimator for point $p_i^l$.

Then we reuse $\alpha_i$ for upsampling coordinates, scene flow. Other methods don't involve point coordinates upsampling and the common process of upsampling feature, and scene flow is performed separately.

**Warping layer.** We use the upsampled scene flow to warp.

$$\left(p_i^w\right)^{l-1} = p_i^{l-1} + \text{up}\left(s_i^l\right) \tag{14}$$

where $\text{up}\left(\right)$ indicates WSA upsampling process.

**Cost volume.** We construct the cost volume between the warped source point $P^w$ and the target point $Q$, which reduces the search area. We adopt patch-to-dilated-patch cost volume to enlarge the receptive field, following RMS-FlowNet(Battrawy et al., 2022).

**Deformation degree module.** First, computing the variables $\delta$ which represent the local structure for $P$ and $P^w$. Second, measuring the difference between $\delta(P)$ and warped $\delta(P^w)$. Note that, it will be inputted into the estimator as the supplement of geometric information.

**Scene flow estimator.** The input to the scene flow estimator consists of five components: the feature $f^{l-1}\left(p_i\right)$ of point cloud $p_i$, cost volume $CV_i$, deformation degree module $\delta_{DD_i}$, the upsampled output feature of estimator in the coarse layer $\text{up}\left(f_e^l\right)$, the upsampled scene flow $\text{up}\left(s^l\right)$. Inspired by DenseNet (Huang et al., 2017), we adopts feature reuse and bypass set in scene flow estimator. The output feature of estimator $f_e^{l-1}\left(p_i\right)$ which represents the flow motion information and the predicted scene flow are as follows.

$$f_e^{l-1}\left(p_i\right) = \text{MLP}\left(\left[f^{l-1}\left(p_i\right), CV_i, \delta_{DD_i}, \ \text{up}\left(f_e^l\left(p_i\right)\right), \text{up}\left(s_i^l\right)\right]\right) \tag{15}$$

$$s_i^{l-1} = FC\left(f_e^{l-1}\left(p_i\right)\right) \tag{16}$$

where $f_e^{l-1}\left(p_i\right)$ denotes the output feature of the estimator. $s_i^{l-1}$ denotes the predicted scene flow. $[\cdot, \cdot]$ indicates concatenation operation. $\text{up}\left(\right)$ indicates WSA upsampling process. FC denotes fully connected layer.

## 3.5 LOSS

The whole system is in a fully-supervised manner by using multi-scale loss $\mathcal{L}_S$ the same as (Wu et al., 2020; Cheng & Ko, 2022; Battrawy et al., 2022). $\mathcal{L}_S$ can be expressed as:

$$\mathcal{L}_S = \sum_{l=0}^{L} \gamma^l \sum_i \left\| \widehat{s}^l - s_{gt}^l \right\|_2 \tag{17}$$

where $\gamma^l$ represents the weight for each pyramid layer $l$. The weights are set as $\gamma^0 = 0.02$, $\gamma^1 = 0.04$, $\gamma^2 = 0.08$, $\gamma^3 = 0.16$, $\gamma^4 = 0.16$. The predicted scene flow is denoted by $\widehat{s}^l$. And $|| * ||_2$ refers to the $L_2$-norm.

Besides, we introduce a deformation degree loss $\mathcal{L}_{DD}$ to maintain geometric invariance. $\mathcal{L}_{DD}$ can be expressed as:

$$\mathcal{L}_{DD} = \sum_{l=0}^{L} \gamma^l \sum_i \left\| \delta_{DD}^l \right\|_2 \tag{18}$$

where $\delta_{DD}^l$ indicates the deformation degree module.

The overall loss $\mathcal{L}_{all}$ comprises $\mathcal{L}_S$ and $\mathcal{L}_{DD}$.

$$\mathcal{L}_{all} = \alpha_S \mathcal{L}_S + \alpha_{DD} \mathcal{L}_{DD} \tag{19}$$

where $\alpha_S = 1.0$ and $\alpha_{DD} = 0.01$ are the weights for each term.

## 4 EXPERIMENTS

### 4.1 DATASETS AND EVALUATION METRICS

**Datasets.** To verify the effectiveness of our method, we use the synthetic dataset FlyingThings3D (Mayer et al., 2016) and real scene dataset KITTI Scene Flow 2015 (Menze et al., 2018) the same as previous methods. FlyingThings3D dataset contains 19,640 pairs in training set and 3,824 pairs in the test set. KITTI dataset contains 200 pairs in training set and 200 pairs in the test set. We follow HPLFlowNet (Gu et al., 2019a) to preprocess data. Due to the disparity of the KITTI test set is not available, we generate a reduced number of 142 pairs of point clouds from the KITTI training set.

**Evaluation Metrics.** For fair comparison, we evaluate the scene flow by following metrics, the same as (Wu et al., 2020; Cheng & Ko, 2022; Battrawy et al., 2022).

● *EPE3D*, *Acc3DS*, *Acc3DR*, *Outliers3D*, *EPE2D*, *Acc2D*.

### 4.2 EXPERIMENTAL SETUP

We conduct experiments on NVIDIA RTX 3090 GPUs. We train on the FlyingThings3D training dataset. The inputs of our network are only two frame point coordinates and the input size is 8192 by randomly sampling. To speed up, the training process is divided into two stages. We first train our model on a quarter of the training set (4910 pairs), then fine-tune on the whole training set. In the first stage, the learning rate is set to 0.001 and the decay rate is 0.7 for every 20 epochs. Pre-training is done for 80 epochs. In the second stage, the learning rate is set to 0.000343 and the decay rate is the same as before. Fine-tuning is done for 160 epochs after loading the pre-trained model. The parameters of Adam optimizer are set to $\beta_1 = 0.9$, $\beta_2 = 0.99$, $weight\_decay = 0.0001$. We evaluate on the FlyingThings3D test dataset to demonstrate the effectiveness. In addition, to verify the generalization capability of our method, we test the model on the KITTI dataset without fine-tuning.

### 4.3 MAIN RESULTS

We compare with the published state-of-the-art methods on the FlyingThings3D (Mayer et al., 2016) and KITTI Scene Flow 2015(Menze et al., 2018) datasets, the quantitative results are shown in Tab. 1. Among the listed methods, only 3DFlow(Wang et al., 2022) evaluates 2048 points, while others evaluate 8192 points' scene flow results.

Table 1: **Quantitative results on FlyingThings3D and KITTI Scene Flow 2015 datasets.** All listed approaches are only trained on FlyingThings3D dataset in a fully-supervised manner. The best results are marked in bold.

| Dataset | Method | EPE3D(m)↓ | Acc3D Strict↑ | Acc3D Relax↑ | Outliers3D↓ | EPE2D↓ | Acc2D↑ |
|---|---|---|---|---|---|---|---|
| FlyingThings3D (Mayer et al., 2016) | PointPWC-Net(Wu et al., 2020) | 0.0588 | 0.7379 | 0.9276 | 0.3424 | 3.2390 | 0.7994 |
| | RMS-FlowNet(Battrawy et al., 2022) | 0.0560 | 0.7920 | 0.9550 | 0.3240 | - | - |
| | HCRF-Flow (Li et al., 2021) | 0.0488 | 0.8337 | 0.9507 | 0.2614 | 2.5652 | 0.8704 |
| | PV-RAFT(Wei et al., 2021) | 0.0461 | 0.8169 | 0.9574 | 0.2924 | - | - |
| | FlowStep3D(Kittenplon et al., 2021) | 0.0455 | 0.8162 | 0.9614 | 0.2165 | - | - |
| | RCP(Gu et al., 2022) | 0.0403 | 0.8567 | 0.9635 | 0.1976 | - | - |
| | Bi-PointFlowNet(Cheng & Ko, 2022) | 0.0280 | 0.9180 | 0.9780 | 0.1430 | 1.5820 | 0.9290 |
| | 3DFlow (Wang et al., 2022) | 0.0281 | 0.9290 | 0.9817 | 0.1458 | 1.5229 | 0.9279 |
| | Ours | **0.0239** | **0.9391** | **0.9821** | **0.1103** | **1.3703** | **0.9358** |
| KITTI (Menze et al., 2018) | PointPWC-Net(Wu et al., 2020) | 0.0694 | 0.7281 | 0.8884 | 0.2648 | 3.0062 | 0.7673 |
| | RMS-FlowNet(Battrawy et al., 2022) | 0.0530 | 0.8180 | 0.9380 | 0.2030 | - | - |
| | HCRF-Flow (Li et al., 2021) | 0.0531 | 0.8631 | 0.9444 | 0.1797 | 2.0700 | 0.8656 |
| | PV-RAFT(Wei et al., 2021) | 0.0560 | 0.8226 | 0.9372 | 0.2163 | - | - |
| | FlowStep3D(Kittenplon et al., 2021) | 0.0546 | 0.8051 | 0.9254 | 0.1492 | - | - |
| | RCP(Gu et al., 2022) | 0.0481 | 0.8491 | 0.9448 | **0.1228** | - | - |
| | Bi-PointFlowNet(Cheng & Ko, 2022) | 0.0300 | 0.9200 | 0.9600 | 0.1410 | 1.0560 | 0.9490 |
| | 3DFlow(Wang et al., 2022) | 0.0309 | 0.9047 | 0.9580 | 0.1612 | 1.1285 | 0.9451 |
| | Ours | **0.0277** | **0.9209** | **0.9613** | 0.1350 | **0.9773** | **0.9574** |

Table 2: **Ablation studies on FlyingThings3D.**"MN": modified network. "DD":deformation degree module. "WSA":weight-sharing aggregation module. "✓" denotes using this module. The best results are marked in bold.

| MN | DD | WSA | EPE3D(m)↓ | Acc3D Strict↑ | Acc3D Relax↑ | Outliers3D↓ | EPE2D↓ | Acc2D↑ |
|---|---|---|---|---|---|---|---|---|
| ✓ | | | 0.0282 | 0.9308 | 0.9799 | 0.1317 | 1.7321 | 0.9273 |
| ✓ | ✓ | | 0.0279 | 0.9264 | 0.9786 | 0.1410 | 1.6202 | 0.9223 |
| ✓ | | ✓ | 0.0243 | 0.9387 | 0.9819 | 0.1162 | 1.3868 | **0.9358** |
| ✓ | ✓ | ✓ | **0.0239** | **0.9391** | **0.9821** | **0.1103** | **1.3703** | 0.9358 |

On FlyingThings3D dataset, our method outperforms prior SOTA work on all evaluation metrics, which proves the effectiveness of our method. Our method surpasses current SOTA method Bi-PointFlowNet (Cheng & Ko, 2022) by $14.6\%$ on EPE3D metric. We surpass HCRF-Flow (Li et al., 2021) which utilizes the direct rigidity constraints by $51.0\%$ on EPE3D metric.

On KITTI Scene Flow 2015 dataset, we evaluate our model without fine-tuning to verify the generalization ability. Our method outperforms prior SOTA method Bi-PointFlowNet (Cheng & Ko, 2022) by $7.6\%$ on EPE3D metric.

In Fig. 4, the visualization results demonstrate the better accuracy of our method than recent SOTA methods(Wang et al., 2022; Cheng & Ko, 2022) on FlyingThings3D and KITTI Scene Flow 2015 datasets. We also show the local details for easy observation. In some challenging areas, we still achieved good results (fewer red points means fewer errors).

## 4.4 ABLATION STUDIES

**Modified network.** As shown in the first row of Tab. 2, the modified network we proposed (MN) outperforms the baseline network PointPWC-Net(Wu et al., 2020) and RMS-FlowNet (Battrawy et al., 2022) by a large margin.

**Deformation degree module.** We compared the results with and without deformation degree module. Equipping the module on the modified network structure can improve the performance. Even in outstanding designs (MN + WSA), there is still performance improvement by using deformation degree module (MN + DD + WSA). Therefore, preserving local structure can obtain more accurate scene flow.

**Weight-sharing aggregation.** Contrast of the results of the first (MN) and third rows (MN + WSA) illustrates that adding a weight-sharing aggregation module can bring $13.8\%$ accuracy improvement on EPE3D metric. It proves the validity of the weight-sharing aggregation constraints which use identical weights for aggregation of point coordinates, scene flow and features. It is consistent with the previous formula derivation conclusion (§ 3.2).

According to the analysis of the experimental results of the above different designs, each module we proposed is effective. The best method design consists of three modules (MN + DD + WSA).

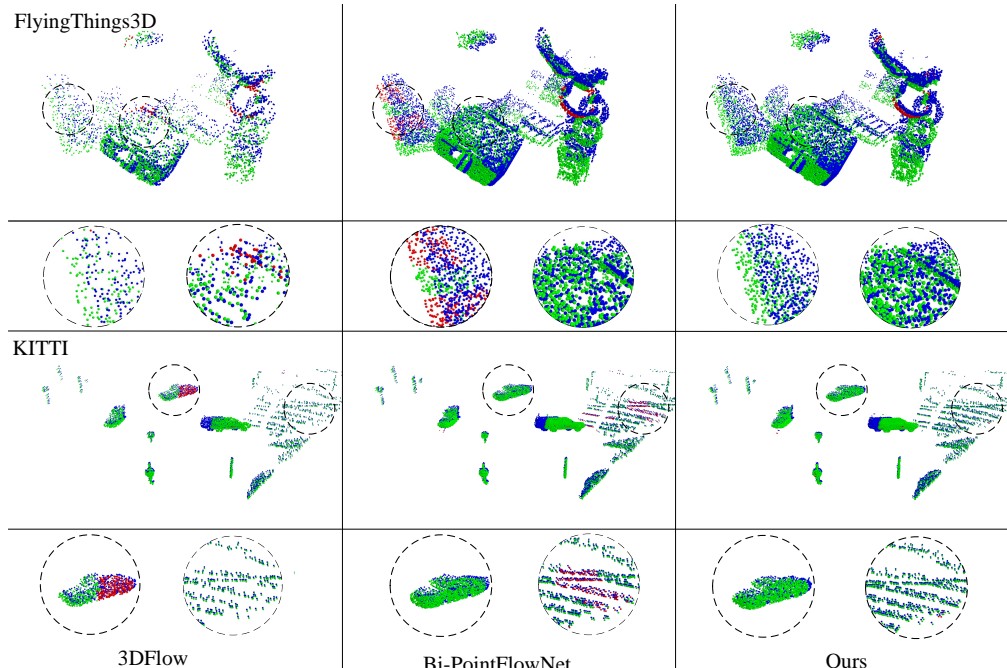

FlyingThings3D

KITTI

3DFlow            Bi-PointFlowNet           Ours

Figure 4: **Visualization Results.** From left to right: 3DFlow, Bi-PointFlowNet and our method. The first two rows are the experimental results on FlyingThings3D dataset. The last two rows are the experimental results on KITTI dataset. Blue represents the source point. Green is the correctly predicted warped source by the *Acc3DS* metric, while red indicates an incorrect prediction.

Table 3: **Universality studies of WSA and DD modules.**

| Bi-PointFlowNet | WSA | DD | EPE3D(m)↓ | Acc3D Strict↑ | Acc3D Relax↑ | Outliers3D↓ | EPE2D↓ | Acc2D↑ |
|---|---|---|---|---|---|---|---|---|
| ✓ | | | 0.0280 | 0.9180 | 0.9780 | 0.1430 | 1.5820 | 0.9290 |
| ✓ | ✓ | | 0.0243 | **0.9425** | 0.9853 | **0.1134** | 1.3178 | **0.9511** |
| ✓ | ✓ | ✓ | **0.0241** | 0.9416 | **0.9856** | 0.1181 | **1.2983** | 0.9504 |

## 4.5 UNIVERSALITY STUDIES

To demonstrate the universality of our method, we integrate the proposed modules (WSA and DD) into current SOTA method Bi-PointFlowNet (Cheng & Ko, 2022). As a result, as illustrated in Tab. 3, we observe a 13.9% improvement on EPE3D metric when our modules were applied to Bi-PointFlowNet.

## 5 CONCLUSION

In this paper, we propose weights-sharing aggregation constraints to employ the rigidity constraints indirectly, which avoids the errors caused by combining with other 3D tasks. Weights-sharing aggregation constraints align the aggregated weights of feature and scene flow with the aggregated weights of point coordinate. We prove the effectiveness of the constraints by formula derivation and experiments. In addition, we further keep the local geometric structure invariance by constructing the deformation degree module, which represents the structural difference in local areas between the source domain and the target domain. It also provides geometric information for the subsequent estimator to obtain more precise scene flow. We modify the coarse-to-fine network and equip it with the module we proposed above. Experiments performed on the FlyingThings3D and KITTI scene flow datasets illustrate the effectiveness and generalization capability of our WSAFlowNet.

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

# A  APPENDIX

**Evaluation Metrics.**

- *EPE3D*: average end point error per point, measured in meters, $\left\| \widehat{S}^l - S_{gt}^l \right\|_2$.

- *Acc3DS*: the fraction of points with *EPE3D* $< 0.05m$ or relative error $< 5\%$.

- *Acc3DR*: the percentage of points where *EPE3D* $< 0.1m$ or relative error $< 10\%$.

- *Outliers3D*: points percentage with *EPE3D* $< 0.05m$ or relative error $< 5\%$. with *EPE3D* $> 0.3m$ or relative error $> 10\%$.

- *EPE2D*: the 2D average end point error, derived from projecting back onto the image plane.

- *Acc2D*: the fraction of points whose *EPE2D* $< 3px$ or relative error $< 5\%$.

