# OpenReview forum: "Exploiting Implicit Rigidity Constraints via Weight-Sharing Aggregation for Scene Flow Estimation from Point Clouds"
_ICLR.cc/2024/Conference — ICLR 2024 Conference Withdrawn Submission_

### Official Review · Reviewer_BuUw · 2023-10-30

**Soundness:** 2 fair
**Presentation:** 3 good
**Contribution:** 2 fair
**Rating:** 3
**Confidence:** 4

**Summary:**

This manuscript addresses the problem of scene flow estimation from point clouds. A weight-sharing
aggregation (WSA) module and a deformation degree (DD) module are proposed to implement implicit rigidity constraints. The resulting algorithm has been evaluated on FlyingThings3D and the KITTI dataset without occlusion.

**Strengths:**

1. The manuscript is overall well written and easy to follow.
2. The proposed method shows good performance on FlyingThings3D and the KITTI dataset
without occlusion.
3. The proposed WSA and DD modules are implemented on both PointPWC-Net and
Bi-PointFlowNet and show a performance improvement.

**Weaknesses:**

1. Important related works are missing. This work is not the first to implement rigidity
constraints to scene flow estimation. See references below.
2. What happens if the objects are deformable, e.g. pedestrians. Does the method perform worse?
It would be good if there was a visual comparison / analysis for such circumstances.
3. How is Eq(7) inferred? Obviously, Eq(7) holds true if f_e() is a linear
function. However, there are usually non-linearities in deep neural networks and Eq(7) needs to be shown.
4. After the upsampling layer, the coordinates and the features are updated. How is the
concatenation with the downsampled features done given the new coordinates?
5. The evaluation is done on Flyingthings3D and KITTI following the preprocessing in
HPLFlowNet. However, in this setting the occlusion points are removed, which is usually
not the case in the real world. How does the proposed method work following the
preprocessing in FlowNet3D with occlusions? And how does it work on a larger scale
autonomous driving dataset such as Waym-Open (https://waymo.com/open/data/motion/)?
6. The deformation degree module seems to show a marginal improvement.

References:

Teed Z, Deng J. Raft-3d: Scene flow using rigid-motion embeddings[C]//CVPR2021.

Dong G, Zhang Y, Li H, et al. Exploiting rigidity constraints for lidar scene flow estimation
[C]//CVPR 2022.

Li R, Zhang C, Lin G, et al. Rigidflow: Self-supervised scene flow learning on point clouds
by local rigidity prior[C]//CVPR2022.

**Questions:**

1. What is the novelty / difference in relation to the references given above?

2.-5. Please refer to weaknesses 2.-5.

---

### Official Review · Reviewer_YnNz · 2023-10-31

**Soundness:** 3 good
**Presentation:** 3 good
**Contribution:** 2 fair
**Rating:** 3
**Confidence:** 4

**Summary:**

The authors propose a weight-sharing aggregation based scene flow estimation network. A weight-sharing aggregation module is proposed to utilize the aggregated weights of point coordinates to aggregate the scene flow and point features for scene flow and feature upsampling. It aims to implicitly encode the rigidity constraints into the upsampling stage. Then a deformation degree module is proposed to preserve the local geometric structures between the source point clouds and the warped target point clouds. Extensive experiments on synthetic FlyingThings3D and real KITTI datasets demonstrate its effectiveness.

**Strengths:**

(1)	Different from that the methods that can explicitly encode the rigidity constraints based on the segmented objects and rigid pose estimation, the proposed method can implicitly encode the rigidity constraints into the upsampling stage with sharing weights.
(2)	The experiments show the  superiority of the proposed method.

**Weaknesses:**

(1)	This paper claims that scene flow and point coordinates can be aggregated with the sharing weights. But I am confused about the operation on aggregating features with the same weights. I suggest that authors could conduct the experiments on upsampling the point features with different aggregated weights. Will the performance decrease a lot with different parameters?
(2)	The authors should verify that the generated scene flow via weight-sharing aggregation really conforms to the rigid constraints.
(3)	Some important details are missing and the descriptions are unclear. For example, this apper lacks a description of how to obtain the aggregated weights. And what does “MN” mean? Does it correspond to the feature pyramid in Section 3.4?
(4)	The ablation study on “DD” module shows that it brings the network a very weak improvement.
(5)	The experimental section lacks the complexity analysis of the proposed method. Please list the inference time and FLOPs of compared methods.

**Questions:**

Please see the weakness section.

---

### Official Review · Reviewer_gEsZ · 2023-11-01

**Soundness:** 3 good
**Presentation:** 3 good
**Contribution:** 2 fair
**Rating:** 6
**Confidence:** 3

**Summary:**

The paper studies scene flow estimation from point clouds, with emphasizing implicit rigidity constraints. Following the design of PointPWC-Net, the core of this approach lies in two primary components: a weight-sharing aggregation method and a deformation degree module. While the former leverages identical weights for the fusion of point coordinates, scene flow, and features to implicitly establish rigidity constraints, the latter emphasizes the preservation of local structures of rigid objects during the scene flow estimation process.

This paper showcases the efficacy of the resultant network, WSAFlowNet, by benchmarking its performance against the popular FlyingThings3D and KITTI datasets.

**Strengths:**

- Overall, the writing of the manuscript is mostly clear and easy to follow. The authors provided various mathematical and experimental evidence to support the feasibility and effectiveness of their approach.
- In feature abstraction, the proposed weight-sharing aggregation module enforces rigidity constraints by using identical weights for aggregating point coordinates, scene flow, and features, without requiring explicit pose estimation or 3D object segmentation.
- The proposed deformation degree module reasonably enhances the local structure consistency by measuring the deformation degree of local structure between reference and warped points, with a KNN.
- Both proposed modules are properly ablated, which provides convincing signals with analysis.

**Weaknesses:**

- Based on the ablations in Table 2, the gain from the deformation degree module looks minor. Indeed the two proposed modules work well in tandem, but more analysis and experimental evidence are needed to verify the necessity of the DD module for maintaining local rigid structure invariance.
- Relevant comparisons on the computation cost of different approaches could be useful for evaluating the practicality of the proposed model for the scene flow estimation task.
- No potential limitations or drawbacks of the proposed method are discussed. Also, failure cases need to be showcased and analyzed if any. Here the paper seems to focus too much on the advantages of the presented method and does not always give the whole picture.
- I'm not super familiar with recent SOTA methods on scene flow estimation from point clouds, so might be biased on this one: the generalization capability of the proposed model was demonstrated by evaluating on KITTI without finetuning, but it would be more convincing to demonstrate comparisons on more recent real-world point cloud data from autonomous driving datasets, such as nuScenes, Argoverse, and Waymo open dataset.

**Questions:**

Please check the suggestions in the Weakness section above to further strengthen the paper.